# Effects of Short-Term High Temperature on Gas Exchange in Kiwifruits (*Actinidia* spp.)

**DOI:** 10.3390/biology11111686

**Published:** 2022-11-21

**Authors:** Dawei Li, Xiaodong Xie, Xiaoying Liu, Chang Cheng, Wen Guo, Caihong Zhong, Arif Atak

**Affiliations:** 1CAS Engineering Laboratory for Kiwifruit Industrial Technology, Wuhan Botanical Garden, Chinese Academy of Sciences, Wuhan 430074, China; 2College of Horticulture, Anhui Agricultural University, West Changjiang Road, Hefei 230036, China; 3Yunnan Key Laboratory of Plant Reproductive Adaptation and Evolutionary Ecology, Institute of Biodiversity, School of Ecology and Environmental Science, Yunnan University, Kunming 650500, China; 4Department of Horticulture, Agriculture Faculty, Uludağ University, Bursa 16059, Turkey

**Keywords:** abiotic stress, photosynthetic rate, high temperature, transpiration rate, kiwifruit

## Abstract

**Simple Summary:**

Climate changes occurring today require detailed research of the effects of increasing temperatures on photosynthesis in different species and cultivars. In addition, it is very important to determine how different Actinidia species growing in the temperate climate zone will respond to global climate change and increasing temperatures. Temperature variability is a crucial determinant of the yield and quality of plants, particularly when high-temperature episodes coincide with their growth and fruit development. Species and cultivars that can adapt to extreme temperatures and are least affected by these temperatures should be determined and used for high-quality and profitable production in the coming years. In this study, the effects of environmental factors on photosynthetic activity and gas exchange in Actinidia species with different ploidy levels were evaluated. Tetraploids showed higher high-temperature resistance, while hexaploids had the highest net photosynthetic rate. Our research showed that the high-temperature tolerance of kiwifruits existed not only in different species but also among cultivars. As a result, it was determined that high temperatures have important effects on the photosynthetic activities of kiwifruit plants with different ploidy levels, and that these effects can significantly change their development according to how they differ among different species/cultivars.

**Abstract:**

Climate changes occurring today require detailed research of the effects of increasing temperatures on photosynthesis in different species and cultivars. Temperature variability is a crucial determinant of the yield and quality of plants, particularly when high-temperature episodes coincide with their growth and fruit development. The effect of high temperature (HT) on higher plants primarily concerns photosynthetic functions, but the sensitivity of photosynthesis to high temperature is not well-understood in kiwifruits. In this study, we designed a new environmental monitoring system to evaluate the effects of environmental factors on the photosynthetic physiology of different kiwifruit species and cultivars. A significant positive correlation was determined between the main photosynthetic indices of kiwifruits, such as transpiration rate and net photosynthetic rate. The net photosynthetic rate of commercial kiwifruit cultivars was strongly inhibited when the temperature exceeded 44.5 °C, and the leaves of kiwifruits were irreversibly damaged when the temperature increased to 52 °C. Kiwifruit cultivars with different ploidy levels (diploid, tetraploid and hexaploid) were found to be sensitive to high temperature, whereas tetraploids had higher HT resistance and hexaploids had the highest net photosynthetic rate. Further research showed that the HT tolerance of kiwifruits existed not only between species but also among cultivars. *A. eriantha* had the highest net photosynthetic rate at more than 44.7 °C, but those of *A. deliciosa* and *A. arguta* declined sharply as the temperature exceeded 43.5 °C. As a result, it was determined that high temperatures have important effects on the photosynthetic activities of kiwifruit plants with different ploidy levels, and that these effects can significantly change their development according to how they differ among different species/cultivars.

## 1. Introduction

Due to climate fluctuations, plants encounter increasing biotic and abiotic stress combinations, severely affecting their growth and yield [1,2]. Resistance to various stress conditions during the growth and development of plants is among the most studied subjects of plant physiology and plant biotechnology today [3,4].

Temperatures have increased over the years due to global warming, and high temperature is becoming a serious threat to crop yield and quality [5,6]. Heat stress is often defined as a rise in the temperature beyond a threshold level for an amount of time sufficient to cause irreversible damage to plant growth and development. In extreme temperatures, serious damage can occur to the cellular structures of plants, and this can cause damage to the cells, causing rapid collapse and sometimes even cell death [7]. The term threshold temperature refers to the average daily temperature value at which a detectable decrease begins during the growth of plants. For example, when the ambient temperature of the tomato plant exceeds 35 °C, many critical physiological processes, such as seed germination, growth, flowering, fruit set and fruit ripening, are adversely affected [8]. However, the threshold temperatures of many crops are unknown, and this causes much inconvenience to crop production [9]. For a sustainable production of the desired quality, it is very important to determine the lower and upper growth threshold temperatures for plant species together with controlled field and also laboratory conditions.

High temperature stress is one of the major abiotic stresses that limits plant productivity in various ways [10]. Terrestrial plants are often subjected to strong temperature changes. These variations can reach temperatures of 40 °C or more in both temperate and hot desert regions. Plants have to cope with changes in their environment with reduced mobility depending on the soil [11]. The current increase in greenhouse gas emissions will result in a 2 °C to 3 °C increase in atmospheric temperature over the next 50 years. At the same time, heat waves and periods of extreme heat will be more frequent and of longer duration. Agricultural production and the functioning of forests will therefore be greatly affected. Modeling studies show that if agricultural adaptation to these new conditions is not achieved, the decrease in crop yield could reach 17% for every 1 °C increase in growing season temperature [12,13]. For this reason, it is important to identify species and varieties that are more resistant to these increasing temperatures and to use them in breeding studies [14].

Although the growth and development of plants are regulated by some biochemical, physiological and molecular processes, photosynthesis yield potential has an especially great effect on them. Suppose one of the important components of photosynthesis (such as CO_2_ reduction pathways, two photosystems, photosynthetic pigments and electron transport system) is damaged. In that case, it can inhibit the photosynthetic mechanism of the plant. [15]. In studies on photosynthesis and its functioning in plants, it is reported that photosynthesis is a very important part of the complex physiological process of plants. This process includes many components (especially the electron transport system, CO_2_ reduction pathways and photosynthetic photosystems). Researchers have also stated that photosystem II (PSII) is one of the most heat-sensitive components in photosynthesis [16,17]. As a result of the negative effects of high temperatures (constant or temporary), especially on the growth and productivity of plants, photosynthesis is reduced and other changes arise, especially in biochemistry and physiology [18]. While heat stresses (especially moderate and severe heat stress) causes a reversible decrease in photosynthesis, increased heat stresses can cause irreversible damage to the photosynthetic process and ultimately inhibit plant growth significantly [19]. Therefore, assessing the effects of heat stress on plant photosynthesis can not only determine the plant threshold temperature but also help us screen heat-resistant germplasm resources.

Kiwifruit (*Actinidia chinensis* var. *deliciosa*) is a subtropical species from the *Actinidiaceae* family that has a crucial economic role due to its edible fruit and rich ascorbic acid content [20]. Kiwifruit has become one of the most important fruits globally, and its production in 2019 reached approximately 4.35 million tons. China is the most important producer, followed by New Zealand, Italy, Iran, Greece and Chile. The planting areas in these countries are mainly located in sub-hotspot areas where the high temperature in summer is one of the important threats to the kiwifruit’s high and stable yield [21]. There are numerous reports on the effect of high temperature on the many components of kiwifruit (*Actinidia* spp.), such as its physiology, flower, fruit composition, vine growth and yield. However, it remains unclear how high temperature affects the physiology of kiwifruits [22]. Specifically, we hypothesize that kiwifruits are susceptible to high temperature given their subtropical origin. Characteristics such as ploidy level and morphological traits may cause different responses of gas exchange to high temperature among species and cultivars. This study aimed to evaluate the high-temperature response of kiwifruit species and cultivars with different ploidies and provide scientific and technological support for breeding high-temperature-resistant kiwifruit cultivars in the future.

## 2. Materials and Methods

### 2.1. Plant Material

Information about the kiwifruit species, cultivars and ploidy level used in this study are given in Figure 1. The kiwifruit plant materials used in the study were obtained from Wuhan Botanical Garden, the Chinese Academy of Sciences, Hubei, China (geographical references: 30.46° N–114.61° E). We certify that our experimental research, field studies and all methods concerning kiwifruit plants were carried out in conformity to the relevant institutional, national and international guidelines and legislation. The study was carried out in a climate-controlled greenhouse environment. LI-6400/XT was used to determine the gas exchange data of the kiwifruit plants in the greenhouse. All the plants were grafted on the same rootstock and planted into seven-gallon pots (38 cm diameter × 40 cm height) containing a 2:1:2 ratio of topsoil, sand and fertilizer. The plants were then irrigated and cultured regularly to maintain identical growth conditions in 2020. After a year of training, 5 healthy and well-established kiwifruit vines with 1.2–1.5 cm trunk diameters were randomly selected from each species/cultivar for photosynthetic measurements.

First, we designed a new environmental monitoring system (Figure 2) to evaluate the effects of environmental factors (Figure 3) on the photosynthetic physiology of kiwifruits of different species and varieties. Second, we designed an artificial climate chamber to evaluate the threshold temperature of the kiwifruits and the effect of high temperature on kiwifruit photosynthesis. Third, we analyzed the response of the kiwifruit cultivars with three ploidy levels (diploid, tetraploid and tetraploid) and commercial species (*A. deliciosa, A. arguta* and *A. eriantha*) to high temperature, and screened those with high temperature resistance (Figure 4, Figure 5 and Figure 6).

### 2.2. Measurement of Growth Conditions

During the experiment, the following data concerning the climatic and soil conditions of the plants were measured: temperature (°C), CO_2_ concentration (ppm), relative humidity (%RH), light intensity (lux), soil pH and electrical conductivity (μS/cm). These data were measured with sensors from the Infwin company, Dalian, China (www.infwin.com.cn, accessed on 1 February 2022). An automatic data collection system was developed, automatically sending environmental data to the computer through a 4G network signal every 15 min. Temperature (range: −40–80 °C, resolution: 0.1 °C, accuracy: +/−0.3 °C) and humidity (Resolution: 0.03% within 0–50%, 1% within 50–100%) data for the soil (0 cm, underground −20 cm and −50 cm) and air (above ground 100 cm, 200 cm and 300 cm) were accurately recorded by probes, whereas soil pH and electrical conductivity (resolution: 10 us/cm within 0–10,000 us/cm), as well as the concentration of carbon dioxide (range: 0–5000 ppm; accuracy: +/−(50 ppm + 5% read value); Resolution: 1 ppm) in the air and the intensity of light (range: 0–200,000 Lux; accuracy: +/−5%; resolution: 0.1 Lux), were also determined by different probes from the Infwin company. The detailed data collection model is shown in Figure 2.

### 2.3. Measurement of Photosynthesis

The portable photosynthesis system (LI-6400XT) is an important measurement tool in the field of ecology. It can measure multiple physiological indices, such as net photosynthetic rate (μmol CO_2_ m^−2^s^−1^), stomatal conductance (mol H_2_O m^−2^s^−1^), intercellular CO_2_ concentration (μmol CO_2_ mol^−1^) and transpiration rate (mmol H_2_O m^−2^s^−1^). To measure the diurnal photosynthetic dynamics, the LED chamber and CO_2_ injection system were used to control the light intensity and CO_2_ concentration in the analyzers. The experiment was carried out on sunny, cloudless and high-temperature days. Kiwifruit plants with good growth conditions and without diseases, insect pests or damage were exposed to a range of temperature conditions from 24 °C to 45 °C during the day. All the measurements were carried out on 15 fully expanded and healthy leaves of each plant using a closed portable photosynthesis system (Model LI-6400; LI-COR, Lincoln, NB, USA).

In our study, the response of the yellow-fleshed tetraploid kiwifruit genotype (*A. chinensis*) to high-temperature stress during photosynthesis was primarily investigated (Figure 4). In the temperature-controlled experiment, two enclosed greenhouses were used. The optimum temperature conditions required for the normal development of kiwifruits were created in ROOM1. In the other room (ROOM2), extremely high-temperature conditions (between 25 °C and 57 °C) were created. In this experiment, other environmental factors, such as light intensity, soil temperature and humidity, were kept under control at optimum conditions. For the other parts of the study, the photosynthesis measurement values of the high-temperature application are given in Figure 5 according to the ploidy level of the cultivars and in Figure 6 based on the species. The sampled kiwifruit plants were grown in an average air relative humidity of 66% and watered every other day in these naturally lit greenhouses. The measurements of the environmental conditions of the greenhouse area during the experiment are shown in Figure 3.

### 2.4. Statistical Analysis

The assay was conducted according to a completely randomized design. The data obtained as a result of the study are given as the mean of 15 replicates (5 samples per replicate with a total of 3 replicates) with ± SE. Analysis of variance (ANOVA) and correlation analysis were performed with the SPSS Statistics 21.0 analysis package (IBM Inc., New York, NY, USA). All variables were log10 transformed prior to the analysis to investigate relative effects and to obtain a better homogeneity of variances. The significance was tested at the 5% level.

## 3. Results

The effects of high temperature on photosynthetic development were investigated using a commercially grown yellow-fleshed tetraploid kiwifruit genotype *(A. chinensis*). Two special rooms were created for normal temperature conditions (ROOM1) and extreme temperature (ROOM2) (Figure 4A). When the results of extreme temperature applications with the yellow-fleshed tetraploid genotype were evaluated, it was observed that especially high temperatures had significant side effects on kiwifruit physiology. The transpiration rate increased very rapidly after the extreme temperatures were applied, as expected. The transpiration rate was 2.5 mmol H_2_O m^−2^s^−1^ at 44.5 °C at 10:30 a.m., it reached 7 mmol H_2_O m^−2^s^−1^ at 52.0 °C at 11:30 a.m., while this value was just 2.5 mmol H_2_O m^−2^s^−1^ in ROOM1. Based on these results, we determined the threshold temperature for respiration of *A. chinensis* as temperatures above 44.5 °C. Irreversible leaf damage occurred at this point, especially when the temperature exceeded 52 °C (Figure 4B). In terms of net photosynthetic rate, although there was a rapid increase in kiwifruits treated with extreme temperature in ROOM2, a very rapid decrease was observed after 10:30 a.m.. When the temperature reached 44.5 °C in ROOM2 at 10:30 a.m., the net photosynthetic rate was 8 μmol CO_2_ m^−2^s^−1^, and this value suddenly decreased to 0 μmol CO_2_ m^−2^s^−1^ at 11:30 a.m.. However, in ROOM1, the photosynthetic rate was found to be 6 μmol CO_2_ m^−2^s^−1^ during the same time interval. In ROOM1, the net photosynthetic rate fluctuations were generally more limited (Figure 4C.) A similar situation to that of the net photosynthetic rate was detected regarding stomatal conductance. While the fluctuation was less in ROOM1, there were much more significant fluctuations in ROOM2. The stomatal conductance and transpiration rate increased sharply when the threshold temperature was exceeded to maintain the vine health of the kiwifruits (Figure 4D). In terms of intercellular CO_2_ concentration values, high temperature has little effect on the intercellular CO_2_ concentration. It was observed that the CO_2_ concentration slightly increased following the application of extreme temperatures (Figure 4E).

The resistance to high temperature of the commercially grown yellow (tetraploid ‘Jintao’), red (diploid ‘Donghong’) and green (hexaploid ‘Jinkui’) fleshy kiwifruit cultivars was also investigated (Figure 5). While photosynthesis was inhibited at temperatures above 39.7 °C in both diploid and tetraploid cultivars, photosynthesis continued to increase up to 43.5 °C in the tetraploid Jintao cultivar. The tetraploid Jintao cultivar was found to have significantly higher temperature resistance, while the hexaploid Jinkui was noted as the cultivar with the highest net photosynthesis rate (15 μmol CO_2_ m^−2^s^−1^) at high temperatures (Figure 5A). It was observed that the rate of transpiration was closely related to the temperature, and when the temperature increased to approximately 44 °C, the transpiration rate reached the maximum level in almost all cultivars. With a temperature decrease, the transpiration rate decreased in all cultivars (Figure 5B). The evaluation of stomatal conductance determined that the hexaploid ‘Jinkui’ cultivar had stronger stomatal conductivity than the other diploid and tetraploid cultivars (Figure 5C). When the data on the intercellular CO_2_ concentrations were examined, a general decrease was observed in all varieties, especially with increasing temperatures, followed by a constant course and subsequent increase with the decrease in temperature (Figure 5D).

The physiological effects of high temperatures on three commercially grown kiwifruit species (*A. deliciosa, A. arguta* and *A. eriantha*) were investigated to determine their high-temperature resistance (Figure 6). The net photosynthetic rate of *A. eriantha* at a high-temperature threshold of 44.7 °C was significantly higher than those of *A. deliciosa* and *A. arguta*. The net photosynthetic rate of *A. deliciosa* at a high temperature of 43.5 °C was 14 μmol CO_2_ m^−2^s^−1^. This was significantly higher than that of *A. eriantha* (10 μmol CO_2_ m^−2^s^−1^) (Figure 6A). The transpiration rates of the three species were affected by high temperatures and had similar variation patterns (Figure 6B). The stomatal conductance of both *A. deliciosa* and *A. arguta* was significantly higher than that of *A. eriantha* during the high-temperature period (Figure 6C). The intercellular CO_2_ concentration values decreased at high temperature in all species, and no significant difference was detected between species (Figure 6D).

The environmental conditions of the greenhouse area during the Figure 5 and Figure 6 experimental periods are given in Figure 3. Temperature in the air (°C) was measured at three different levels (100 cm, 200 cm and 300 cm). While the temperature was approximately 27 °C at approximately 06:00 in the morning, it increased rapidly afterwards and reached 45 °C levels by 11:00 a.m., especially at 200 cm and 300 cm. This temperature level continued until 16:00 p.m. and then started to decrease (Figure 3A). It was determined that the air temperature (°C) value in the soil differed depending on the depth of the soil (0 cm, −20 cm and −50 cm). While the temperature on the top surface of the soil (0 cm) was 28 °C at 06:00 a.m., it increased rapidly and reached 37 °C at 14:00 p.m.. After that, it dropped a few degrees every hour. However, temperatures at soil depths of 20 cm and 50 cm followed a much more stable course and did not change much (Figure 3B). The air relative moisture content (%RH) decreased rapidly inversely with air temperature. The air relative moisture content was high in the morning but then decreased very rapidly with the warming of the air (Figure 3C). The carbon dioxide (ppm) content was similar to the air relative moisture content. The carbon dioxide content, which was high in the morning, dropped rapidly as the air warmed up and remained almost constant between 09:00 a.m. and 17:00 p.m.. The soil relative moisture content (%RH) was similarly investigated at three different depths of the soil (0 cm, −20 cm and −50 cm). The relative moisture content of the top surface of the soil (0 cm) followed a fairly constant course and differed by only 1–2% during the day. While there was a similar difference at a soil depth of 20 cm, this change was up to 3% at a soil depth of 50 cm (Figure 3D). With the increase in temperatures during the day, a very limited decrease was observed in the relative moisture content of the soil. The vapor pressure deficit (VPD) increased rapidly with the increasing air temperature and decreasing relative humidity in the morning. The VPD reached its maximum values (2.88 KPa, 4.54 KPa and 4.04 KPa) and then started to decrease (Figure 3E). The illumination intensity (lux) values increased rapidly between 06:00 a.m. and 12:00 p.m., and later decreased at the same rate until 18:00 p.m. (Figure 3F). The pH value of the soil at a depth of 20 cm decreased between 06:00 a.m. and 11:00 a.m., but then increased until 18:00 p.m..

## 4. Discussion

In this study, the changes in the different photosynthetic factors of different kiwifruit species and cultivars in different greenhouse conditions were compared, and their correlations with each other and environmental factors were also investigated. In particular, the changes after an important abiotic stress factor, such as high temperature, were compared.

Sharma et al. [23] reported that abiotic stresses in plants have negative effects, especially on photosystem performance, chlorophyll biosynthesis, gas exchange parameters and electron transport mechanisms, and that this can seriously reduce the photosynthetic efficiency of plants. This study conducted with different kiwifruit species and cultivars determined that increasing temperatures increased the intercellular CO_2_ concentration, as reported in other studies. In addition, very sharp drops occurred in ROOM2 after extreme temperature application. Different researchers have reported that grapes in particular (*Vitis* spp.), like other woody plants, have an internal adaptive mechanism to combat heat stress. Additionally, recent progress in molecular biology has uncovered the major stress response pathways in plants and has broadened the view of abiotic stress responses and plant tolerance [24,25].

Buwalda et al. [26], in their study with *A. deliciosa,* also stated that there was a decrease in the net CO_2_ assimilation rate with increasing temperature (between 8–24 °C) during the day, similar to the results of our study. In a previous study on a grapevine (*Vitis vinifera*) at growth conditions exceeding 40 °C, Soar et al. [27] found that the upregulation of stomatal conductance and the transpiration rate may be adaptive at the expense of transpiration efficiency, which provides evidence to support our study.

Kiwifruit plants (*Actinidia* spp.) have physiologically low stomatal regulation and require a frequent irrigation schedule to maintain adequate plant water status [28]. In addition to climate and soil characteristics, the physiology and morphology of the plant have a significant effect on the determination of the water requirements of the cultivars belonging to the *Actinidia* species [29]. The kiwifruit plant is susceptible to stresses caused by high temperature and humidity, so the amount of water and its management must be adjusted very carefully as these factors significantly affect photosynthesis and fruit development [17]. The high positive correlation between the transpiration rate and stomatal conductance in this study illustrates the sensitivity of the kiwifruit plant.

Judd et al. [30] reported that water loss continued even under water-deficient conditions due to the insufficient stomatal control of kiwifruits. In very low soil moisture conditions, the kiwifruit plant is susceptible to high temperatures and drought due to its low stomatal conductivity. Buwalda et al. [26] reported that decreased photosynthesis of kiwifruit plants in field conditions during the day was highly associated with stomatal closure. In the studies carried out so far, drought stress has been studied mostly in the *A. deliciosa* species among *Actinidia* species, and a very limited study has been done with other economic *Actinidia* species, *A. arguta* and *A. eriantha*. Korean researchers compared kiwifruit species in a study and found that *A. arguta* and *A. eriantha* were more tolerant of drought conditions than *A. chinensis* and *A. deliciosa* [31]. This study clearly shows that the *A. arguta* and *A. eriantha* species differ from the *A. deliciosa* species, especially regarding stomatal conductance and intercellular CO_2_ concentration. As kiwifruit plants showed similar decreases in stomatal conductance and photosynthetic rates under previous drought studies as they did in our high temperature experiments, these would be two important environmental factors which may inhibit the growth of kiwifruits.

In another study, Zhang et al. [32] also demonstrated that the root activity of *A. chinensis* cv. ‘Hongyang’ increased significantly in mild and moderate drought periods and rapidly declined in severe drought periods; the leaf net photosynthetic rate, transpiration rate, stomatal conductance and photochemical efficiency decreased significantly with increasing drought, while the intercellular CO_2_ concentration increased significantly. Researchers have mentioned that the ‘Hongyang’ cultivar gains drought resistance by decreasing the photosynthesis rate, especially during mild and moderate drought periods. Parallel to these findings, our study observed that the photosynthetic activity values were quite different with increasing temperatures among the cultivars, which showed that there might be a difference in resistance to high temperature, and therefore to drought, between cultivars.

Photosynthesis is among the plant cell functions that are highly sensitive to high-temperature stress and are often inhibited before other cell functions are impaired. High-temperature stress has physiological, biochemical and molecular effects on the photosynthesis process. Researchers have reported sensitivity at different rates at high temperatures, depending on the species, cultivar and ploidy level, in their studies on different species. It has also been reported that the effect of high temperatures is not only significant in the first plant development stages but also in different developmental periods [33]. In studies on the adaptation mechanism of plants to heat stress, it is explained that plants avoid excessive heat with the help of a series of strategies or by the use of tolerance mechanisms [5]. When plants are in high-temperature conditions, they exhibit some short-term avoidance or acclimation mechanisms, such as transpiration, leaf orientation or alteration of membrane lipid compositions. In addition, reducing water loss by closing stomata and increasing xylem vessels are some common protection methods deployed by plants against heat-induced stress [34]. This situation is not very different in the kiwifruit plant. Under increased temperature conditions, the kiwifruit plant tries to protect itself from the negative effects of extreme temperatures by increasing the rate of transpiration and its stomatal conductance value. Yan et al. [18] researched the gas and chlorophyll exchange in leaves of sorghum as a result of high temperature, obtaining similar results to ours. They did not detect any change in the photosystem II performance index and photosynthetic rate based on absorption during 1 h at 40 °C. However, they reported that the transpiration rate increased significantly, a self-protective response used by the plant to dissipate the increased heat.

Evans [35] and Cornic [11] have reported that leaves are an important converter that converts solar energy to chemical energy, and also that leaves need a permanent cooling system like any energy converter. During the period of photosynthesis, CO_2_ is absorbed while O2 is released through the stomatal opening (ostiol). For each absorbed CO_2_ molecule, 50–300 molecules of water are transpired from the leaves, depending on the plant type [36]. The water vapour formed as a result of the transpiration of these leaves mainly passes through the ostiole and epidermis. Transpiration allows the continuous cooling of the leaves. In woody lianas, such as some grapevines and kiwifruit plants, leaves can adjust their functioning to cope with high temperature through a high transpiration rate [37,38]. The CO_2_ assimilation of an intact leaf can be rapidly reversed, depending on the temperature range. In other words, CO_2_ assimilation, which increases at optimum temperatures, may decrease rapidly at high temperatures. With a rising temperature, the increased VPD is also an important factor that enhances the water loss from leaves, which can lead to the gradual closing of stomata and a subsequent reduction in transpiration [39]. In our study with different kiwifruit species and cultivars, although the ratio varied according to the species and variety, decreases were clearly observed with increasing temperature, especially in terms of net photosynthesis rate and transpiration rate. This reduction was much more dramatic under extreme temperature applications. Kiwifruit plants try to reduce the effect of increasing temperatures to a minimum level by activating their protective mechanisms at increasing temperatures. Plants can modify their physiology to different ambient temperatures to offset otherwise detrimental effects of temperature fluctuations. Plants can modulate their photosynthesis rates when ambient temperatures are between 10 °C and 35 °C, while their photosynthetic system is damaged by fluctuations below or above these temperatures [40]. This ability to modulate development and physiology in different thermal environments involves signalling mechanisms. Monte et al. [41] have described a new signalling mechanism that protects plants from high temperature stress. OPDA and dn-OPDA activate plant thermotolerance genes in a COI1-independent manner, and treatment with these oxylipins protects plants against heat stress.

Additionally, Zhao et al. [42] explained this situation in their study with cucumber leaves as follows: the reduction in net photosynthesis rate due to the 36 °C high-temperature stress suggests that a large amount of active oxygen species inhibits photosynthesis and damages the mechanism of photosynthesis.

Liu et al. [43], in their study with different kiwi cultivars, reported that the cultivars had high flexibility and adaptation capacity to high-light environments, and that there were significant differences between cultivars. This difference was also seen in terms of photosynthetic activity in our study with different kiwifruit species and cultivars under high-temperature conditions.

## 5. Conclusions

Knowing the reactions of different species to changing and especially increasing temperatures is very important in order to benefit from them in the future. Extreme temperatures, especially with global warming, will become a larger abiotic stress problem in kiwifruit plants, as in many plants, in the coming years. Finding species that are more tolerant of temperature extremes is directly related to better understanding how the greenhouse environment as a whole changes with a forced increase in air temperature, and how changes in vapor pressure affect gas exchange properties, particularly those of stomata. As in this study, determining the photosynthetic activities of different kiwifruit species and cultivars under increasing temperatures will not only guide future breeding studies but will also allow the use of resistant species/cultivars in production. In this context, it is important to determine those resistant to extreme temperatures by carrying out these studies in more species and cultivars.

## Figures and Tables

**Figure 1 biology-11-01686-f001:**
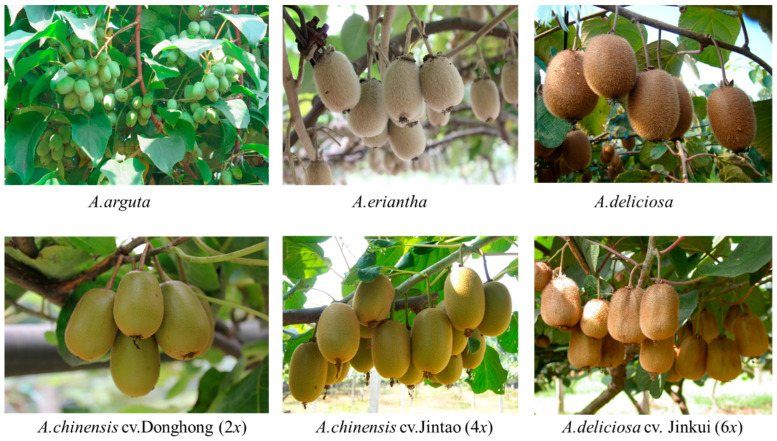
Kiwifruit experimental material for photosynthesis measurement.

**Figure 2 biology-11-01686-f002:**
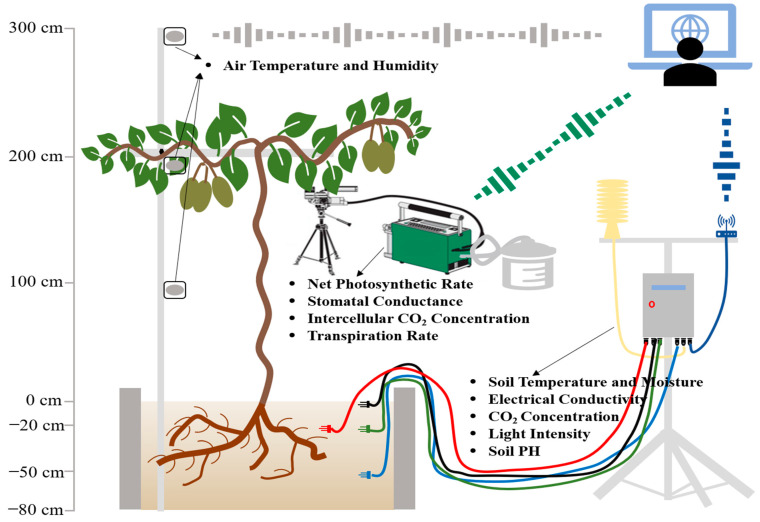
Measuring gas exchange with the LI-6400/XT and environmental detection system.

**Figure 3 biology-11-01686-f003:**
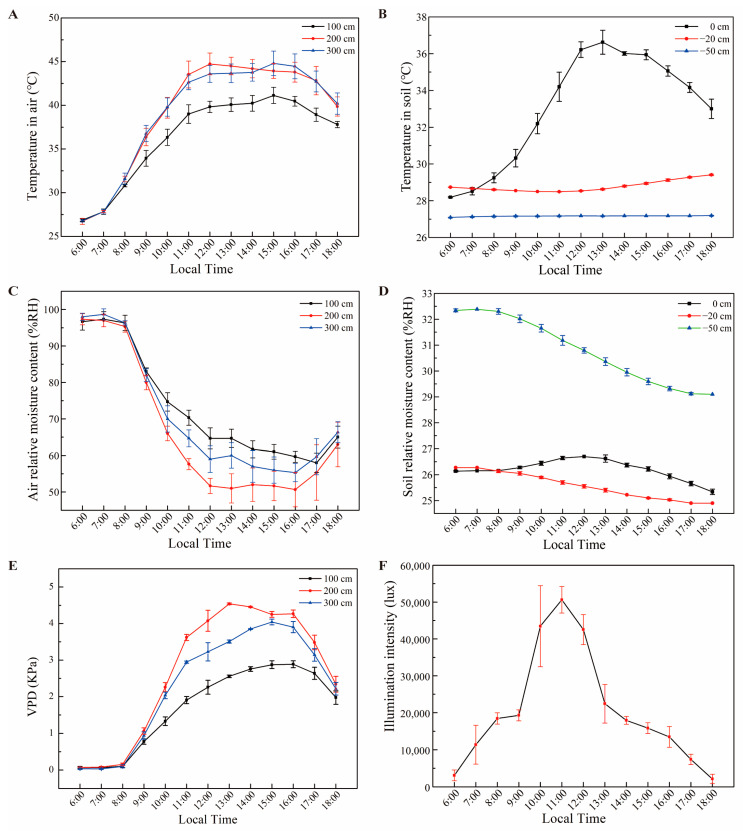
Environmental conditions of the greenhouse area during the experimental period. Temperature and relative moisture content (% RH) in the air and soil are shown, respectively (**A**–**D**). VPD and photon flux densities during the daytime are also shown (**E**,**F**).

**Figure 4 biology-11-01686-f004:**
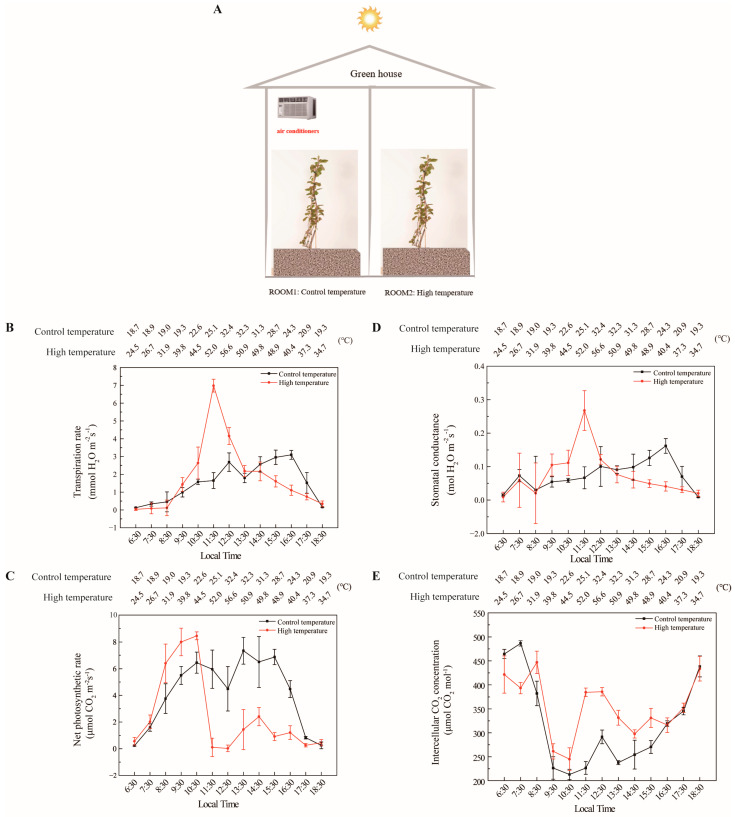
Effect of high-temperature stress on photosynthesis of yellow-fleshed tetraploid kiwifruit genotype (*A.chinensis).* Greenhouse experimental rooms (**A**), transpiration rate (**B**), net photosynthetic rate (**C**), stomatal conductance (**D**) and intercellular CO_2_ concentration (**E**).

**Figure 5 biology-11-01686-f005:**
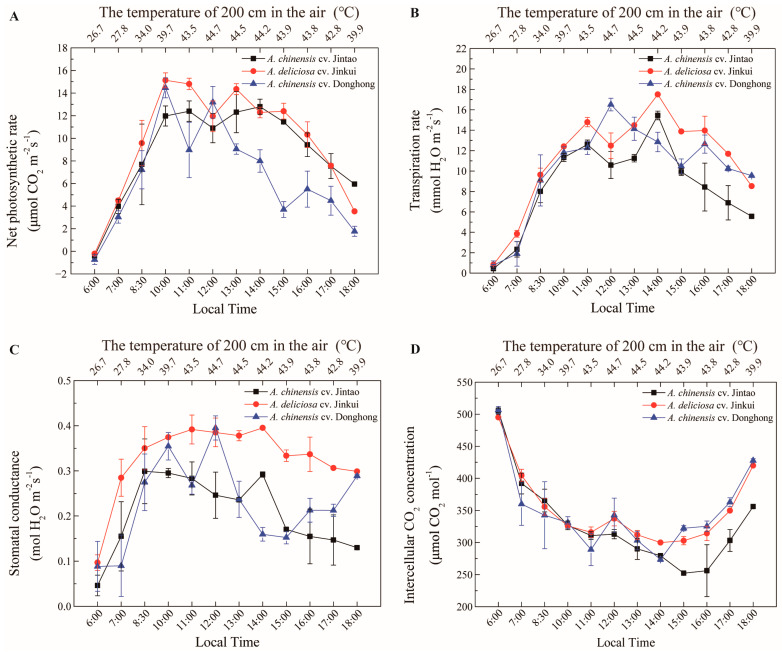
Net photosynthetic rate (**A**) (μmol CO_2_ m^−2^s^−1^), transpiration rate (**B**) (mmol H_2_O m^−2^s^−1^), stomatal conductance (**C**) (mol H_2_O m^−2^s^−1^) and intercellular CO_2_ concentration (**D**) (μmol CO_2_ mol^−1^) changes of diploid, tetraploid and hexaploid kiwifruit cultivars at high temperatures.

**Figure 6 biology-11-01686-f006:**
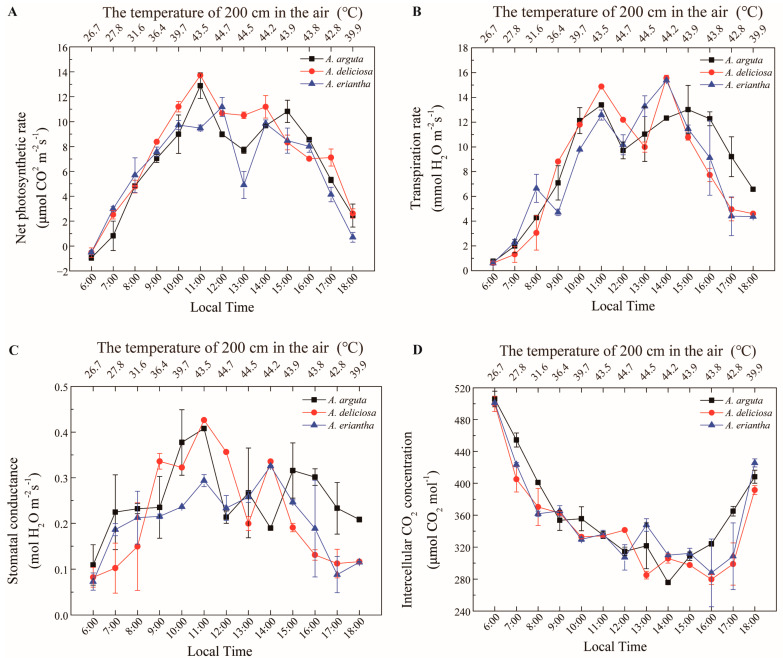
Net photosynthetic rate (**A**) (μmol CO_2_ m^−2^s^−1^), transpiration rate (**B**) (mmol H_2_O m^−2^s^−1^), stomatal conductance (**C**) (mol H_2_O m^−2^s^−1^) and intercellular CO_2_ concentration (**D**) (μmol CO_2_ mol^−1^) changes of diploid, tetraploid and hexaploid kiwifruit cultivars at high temperatures.

## Data Availability

The data supporting the findings of this study are all shown in the figures and are available from the corresponding author upon reasonable request.

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
