# Peer review of "Effects of Short-Term High Temperature on Gas Exchange in Kiwifruits (Actinidia spp.)"

_biology, 2022, doi:10.3390/biology11111686_

Round 1

Reviewer 1 Report (Previous Reviewer 1)

The manuscript looks much better now and ready for publication. 

Author Response

Many thanks to the 1st reviewer for accepting the revised version of our manuscript.

Reviewer 2 Report (Previous Reviewer 2)

Authors have not addressed my concern which was "insufficient data". No new data has been included as I recommended. So, my decision stands same as on initial version. 

Author Response

Answers to Reviewer 2
The second reviewer reported that he did not find the first correction sufficient and insisted on his first comments.
We rearranged our manuscript in accordance with the first comments of the second reviewer. Since his requests were parallel to the third referee, we made the desired changes on the manuscript instead of writing the changes made one by one here. In the second revised article, we have shown all these changes in red. Since we made all the corrections on the second revised text in red, all the corrections requested by the second referee in the first revision will be able to follow from there.
We also removed the correlation table, which was the subject of criticism, and made corrections in the measurement of growth conditions section. After all these changes, we believe that the second reviewer will accept our second revision. Thank you once again for their detailed review.

Reviewer 3 Report (Previous Reviewer 3)

see attached. 

Author Response

After the extensive revision requested by reviewer 3, we hope that reviewer will accept our second time revised manuscript. I'm also adding the answers we gave for the 3rd reviewer here. Many thnaks for your valuable comments.

This manuscript is a resubmission of an earlier submission. The following is a list of the peer review reports and author responses from that submission.

Round 1

Reviewer 1 Report

The authors have worked on a very important fruit that is widely consumed, and a very important climatic factor that has adverse effect on Kiwifruit and crop production in general. However, in order to reach the publication standard, the manuscript needs to be improved as follows: 

The English grammar and sentence framing must be improved throughout the manuscript. A professional English editing is advised. For eg. line 44, 55-56, 59-61, 369, etc. 

Introduction: 

Introduction is short and needs to be elaborated more. 

Line 82: Please correct/elaborate/modify the statement. There are numerous reports on the effect of high temperature on Kiwifruit physiology - flower, fruit composition, fruit and vine growth, yield, etc. 

Materials: 

line 158-159: please mention as technical and biological replicates. 

line 135-136: to my understanding, the environmental conditions shown in figure 7, is a representation of one single day. However, the experiments were performed on several days using one LI-6400. Therefore, graphs showing mean with statistics of the environmental conditions from all the experimental days will be the best. 

line 138-140: to my understanding, all 15 replicates (technical + biological) were done on a single plant of each species. Although measurements were taken on different days, any experimental interpretation/deduction is hard to derive unless the experiment is done on a population or at least 3 biological samples (replicates) (which essentially means 3 different plants). 

Vapor Pressure Deficit (VPD) is an important parameter in this study and should be considered. 

Statistics: Which test is performed to analyse the statistical differences between the means of different groups? 

Results: 

The effects of VPD on the photosynthetic physiology should be tested. 
line 176-177: It is very confusing as which room is controlled low temperature and which room is extreme high temperature. From the figure it looks like that room 1 is air conditioned and with controlled temperature, while room 2 is with high extreme temperature. Please maintain similar explanation throughout the manuscript.    Figures and legends: Some figures are not marked as (A), (B), etc. Some figures are overlapped on top of the other. The legends should describe each and every figure (A, B, C,...etc). Each legend should describe the statistics such as technical and biological reps, SE, p-values, etc. No statistical test is shown to analyse the differences between the groups.    Discussion and conclusion:    The effect of VPD on gas exchange, and thus on the photosynthetic performance should also be discussed.   The discussion and conclusion will change enormously once the above mentioned comments will be taken care of. However, the conclusion is too short and should be elaborated more. 

Reviewer 2 Report

Topic of research is interesting but I have some concerns on the current study.

Data looks insufficient for publication.  Authors must provide more data to strengthen their study. It would be interesting if authors analyze metabolite profile using metabolomics. 

Introduction: hypothesis is weak and a strong rationale is needed to justify the need of current investigation.

Results: these is repetition of results in the writing section.

Discussion: needs to be improved. focus on the molecular mechanisms behind current results.

Language: please improve it and avoid repetition.

Reviewer 3 Report

Review

Effects of short-term high temperature on photosynthesis in Kiwifruit (Actinidia spp.)

By Li et al.

General

This paper attempts to determine the high temperature threshold for several Actinidia species by raising air temperature in some sort of controlled growth facility over the day and measuring gas exchange as the temperature increased and then subsequently decreased.

Aside from the complete lack of description of the growth conditions, there is a flaw in the experimental design. The authors seem to assume that temperature is the only variable changing during the day in the experimental treatment system. Fortunately, the authors do supply Fig. 7 which shows other changes in the treatment conditions. Most notably, the decreasing RH in parallel with the rising temperature indicated the vapour pressure deficit (air) was increasing steeply such that at 45oC, my calculated VPD was 4.2 – 4.6 kPa. Why is this important? Simply because stomatal conductance is inversely and strongly correlated with VPD and hence many changes in stomatal conductance cannot be ascribed directly to the high temperature given the rapid increase in VPD during the day. These two factors, stomatal conductance and VPD, both control transpiration so there is doubt about a direct effect of high temperature on transpiration. Similarly, it was not possible to be sure that the changes in photosynthesis during the temperature rise were attributable to increasing temperature or to stomatal closure from the increasing moisture deficit.  This flaw become magnified if the leaf – air VPD was considered.

On other aspects of the growth conditions as documented in Fig. 7, it was shocking to see light intensity measured in the outdated and no longer accepted Lux whereas it should be measured as photon flux density with a quantum sensor. Furthermore, the very high CO2 in the growth facility in early morning suggested there was poor ventilation in the facility, at least up to 9 am. This should be explained

On the question of threshold temperatures, it does not appear that the authors appreciate that these thresholds very much depend on growth conditions. There is much evidence that grapevines are known to perform adequately at growth conditions exceeding 40oCwhen grown in such conditions, if the authors would like to look for the evidence. There is much sense to this as both Vitis and Actinidia species are lianas and hence have overlapping morphology and physiology and hence more appropriate for comparison instead of sorghum and maize.

Specific comments

L66 Not always in PSII most sensitive – again see grapevine data

L69 Morphology is not dependent on photosynthesis.

L94 I really doubt the LiCor gas exchange system can measure photorespiration

L112 New section ‘Growth conditions’

L115 get rid of Lux

L126 remove photorespiration from the legend, gas exchange is what the Li6400 measures

L131 – 135 use super and subscripts as in the figures eg photosynthesis (µmol m-2 s-1)

L138 how did the growth facility increase in temperature – what mechanism?

L145 Details of all aspects of these enclosed greenhouses

L148 it was imperative that the RH should not have been kept at optimum conditions because as stated above this has lead to a very large atmospheric moisture deficit in the greenhouse.

L180 – 184 this description of changes in transpiration are the phenomenon but not an explanation of why these changes occurred.

L185 what visible leaf damage occurred? How was this detected and what aspects were evident?

L198 not sure of the value of this complex figure.

L 202 – 215 you must mention VPD in this discussion on stomatal conductance and transpiration.

L255 you must describe what instruments were used to measure these environmental conditions.

L 398 there were no measurements of plant growth and crop yield in your study.

L 401 – all the discussion on correlations between stomatal conductance, transpiration and photosynthesis are just hand waving as I have already indicated it very doubtful at gs and E were even affected by the high temperatures.

L420 what relevance is the reference on high CO2 effects to your study?

L449 I think you mean photosynthetic capacity

L 454 how so that your results will guide breeding studies. To find species more tolerant of extreme temperatures, you will need a better understanding of how the greenhouse environment as a whole changes with a forced increase in air temperature and more comprehension of how changes in vapour pressure effects gas exchange attributes, especially stomata.